# Hydroxytyrosol Recovers SARS-CoV-2-PLpro-Dependent Impairment of Interferon Related Genes in Polarized Human Airway, Intestinal and Liver Epithelial Cells

**DOI:** 10.3390/antiox11081466

**Published:** 2022-07-27

**Authors:** Annalisa Crudele, Antonella Smeriglio, Mariarosaria Ingegneri, Nadia Panera, Marzia Bianchi, Maria Rita Braghini, Anna Pastore, Valeria Tocco, Rita Carsetti, Salvatore Zaffina, Anna Alisi, Domenico Trombetta

**Affiliations:** 1Research Unit of Molecular Genetics of Complex Phenotypes, Bambino Gesù Children’s Hospital, IRCCS, 00146 Rome, Italy; annalisa.crudele@opbg.net (A.C.); nadia.panera@opbg.net (N.P.); marzia.bianchi@opbg.net (M.B.); mariarita.braghini@opbg.net (M.R.B.); 2Department of Chemical, Biological, Pharmaceutical and Environmental Sciences, University of Messina, Viale Ferdinando Stagno d’Alcontres 31, 98166 Messina, Italy; antonella.smeriglio@unime.it (A.S.); mariarosaria.ingegneri@unime.it (M.I.); domenico.trombetta@unime.it (D.T.); 3Research Unit of Diagnostical and Management Innovations, Bambino Gesù Children’s Hospital, IRCCS, 00146 Rome, Italy; anna.pastore@opbg.net; 4FACS Unit, Bambino Gesù Children’s Hospital, IRCCS, 00146 Rome, Italy; valeria.tocco@opbg.net; 5B Cell Research Unit, Bambino Gesù Children’s Hospital, IRCCS, 00146 Rome, Italy; rita.carsetti@opbg.net; 6Occupational Medicine/Health Technology Assessment and Safety Research Unit, Clinical-Technological Innovations Research Area, Bambino Gesù Children’s Hospital, IRCCS, 00146 Rome, Italy; salvatore.zaffina@opbg.net

**Keywords:** COVID-19, long-COVID, natural antioxidant, cytokines, interferon, oxidative stress

## Abstract

The SARS-CoV-2 pandemic has caused approximately 6.3 million deaths, mainly due to the acute respiratory distress syndrome or multi-organ failure that characterizes COVID-19 acute disease. Post-acute COVID-19 syndrome, also known as long-COVID, is a condition characterized by a complex of symptoms that affects 10–20% of the individuals who have recovered from the infection. Scientific and clinical evidence demonstrates that long-COVID can develop in both adults and children. It has been hypothesized that multi-organ effects of long-COVID could be associated with the persistence of virus RNA/proteins in host cells, but the real mechanism remains to be elucidated. Therefore, we sought to determine the effects of the exogenous expression of the papain-like protease (PLpro) domain of the non-structural protein (NSP3) of SARS-CoV-2 in polarized human airway (Calu-3), intestinal (Caco-2), and liver (HepG2) epithelial cells, and to evaluate the ability of the natural antioxidant hydroxytyrosol (HXT) in neutralizing these effects. Our results demonstrated that PLpro was able to induce a cascade of inflammatory genes and proteins (mainly associated with the interferon pathway) and increase the apoptotic rate and expression of several oxidative stress markers in all evaluated epithelial cells. Noteably, the treatment with 10 μM HXT reverted PL-pro-dependent effects almost completely. This study provides the first evidence that SARS-CoV-2 PLpro remaining in host cells after viral clearance may contribute to the pathogenetic mechanisms of long-COVID. These effects may be counteracted by natural antioxidants. Further clinical and experimental studies are necessary to confirm this hypothesis.

## 1. Introduction

Severe acute respiratory syndrome coronavirus 2 (SARS-CoV-2) was discovered in January 2020 in Wuhan, China, during the recent epidemic of pneumonia. The virus has spread rapidly throughout the world causing the COVID-19 pandemic [1,2].

The genome of SARS-CoV-2 is about 30 kb in size, contains 14 open reading frames (ORFs) and encodes 29 viral proteins [3]. The first part of SARS-CoV-2 genomic RNA encodes two overlapping polyproteins: pp1a and pp1ab, which, in turn, are digested by two viral proteases, a papain-like protease (PLpro) and a 3C-like protease (3CLpro), to generate 16 non-structural proteins (NSPs). The NSPs assemble into the replication and transcription complex (RTC) to initiate viral RNA replication and transcription [4]. Among the 16 NSPs encoded by SARS CoV-2, NSP3 plays an important role in the viral life cycle because it is essential for RTC formation. NSP3 also functions as a protease; encoded within NSP3 is the PLpro, which cleaves between NSP1-NSP2, NSP2-NSP3 and NSP3-NSP4 to release NSP1, NSP2 and NSP3 from the viral polypeptide [5]. The second part of the viral genome encodes for spike (S) protein, membrane (M) protein, nucleocapsid (N) protein, and envelope (E) protein, which are essential for virion assembly and for suppression of the host immune response [3]. Briefly, the S protein, present on the surface of the virion, binds to the host receptor, angiotensin-converting enzyme 2 (ACE2), which is expressed on the surface of alveolar epithelial type II, renal, cardiac, intestinal, endothelial, and brain cells. After binding, the virus enters the cell to start its life cycle [6,7]. Recently, Wanner et al. provided strong evidence for SARS-CoV-2 liver tropism [8].

Similar to other respiratory viruses, SARS-CoV-2 transmission occurs mainly through the respiratory route [9]. Most patients are asymptomatic or only show mild to moderate symptoms. However, some patients develop respiratory failure, acute respiratory distress syndrome (ARDS) or multi-organ failure [10]. ARDS-positive patients exhibit elevated levels of pro-inflammatory cytokines compared to patients with uncomplicated SARS-CoV-2 infection. The cytokine storm is the result of an uncontrolled immune response with systemic inflammation and hemodynamic instability and is characterized by the abundance of pro-inflammatory cytokines including interleukin (IL)-1, IL-6, IL-18, interferon-γ (IFN- γ) and tumor necrosis factor- α (TNF-α) [11]. Several lines of scientific and clinical evidence have highlighted that many children and adults may manifest subacute and long-term effects after SARS-CoV-2 acute infection, which can affect multiple organ systems [12,13]. This syndrome is known as long-COVID. The World Health Organization (WHO) and global experts went to work to investigate and define the health challenges and implications of the long-COVID condition [14,15]. Lines of evidence suggest that systemic and chronic tissue inflammation, autoimmunity and viral protein/RNA persistence might contribute to the pathogenesis of long-COVID [15,16,17]. PLpro, which is implicated in post-translational modifications of host proteins, thus eliciting the release of interferon-stimulated gene 15 (ISG15) and the consequent exaggerated secretion of multiple pro-inflammatory cytokines and chemokines, could be one of the viral antigens persisting in long-COVID [17]. Moreover, in this scenario, increased oxidative stress seems to play a crucial pathogenetic role [18]. Therefore, a natural antioxidant and anti-inflammatory drugs could be protective factors that may reduce long-COVID symptoms and its public health implications [15].

Plant polyphenols are naturally derived compounds that show anti-viral activity against various pathogenic viruses [19]. Among these compounds, hydroxytyrosol (HXT), a molecule present in olive extracts and olive oil, has several properties including antioxidant, anti-inflammatory, anti-cancer, anti-microbial and anti-viral [20,21]. Hence, HXT could be, as a natural antioxidant, a therapeutic approach to reduce potential future detrimental effects of the long-COVID.

Here, we aimed to evaluate the pro-inflammatory effects of the expression of SARS-CoV-2 PLpro/NSP3 protein in a polarized human airway, intestinal and liver epithelial cells, and next, we evaluated whether the treatment with HXT is able to reduce these effects. 

## 2. Materials and Methods

### 2.1. Cells 

The Calu-3, Caco-2 and HepG2 cells were purchased from American Type Culture Collection (Manassas, VA, USA) that provided certificated authentication. Cells were screened for possible Mycoplasma contamination by using Venor GeM Advance Mycoplasma Detection KIT (Minerva Biolabs GmbH, Berlin, Germany). All the experiments were performed only in Mycoplasma-free cells. 

Calu-3 and Caco-2 were grown as a monolayer in Dulbecco’s modified Eagle’s medium (DMEM)/F-12 GlutaMAX™ supplemented with 10% fetal bovine serum, 2 mM L-Glutamine, 10,000 U/mL penicillin and 10 mg/mL streptomycin at 37 °C under 5% CO_2_ in a 95% humidified atmosphere. Concurrently, HepG2 was grown as a monolayer in DMEM supplemented with 10% fetal bovine serum, 2 mM L-Glutamine, 10,000 U/mL penicillin and 10 mg/mL streptomycin at 37 °C under 5% CO_2_ in a 95% humidified atmosphere.

### 2.2. Transfection and Treatment 

Plasmid transfections in Calu-3, Caco-2 and HepG2 cells were performed by using Lipofectamine2000 (Thermo Fisher Scientific-Invitrogen, Waltham, MA, USA) following the manufacturer’s instructions. In detail, cells were seeded at a density of 300,000 cells/well in a 6-wells plate and were transfected with 2 μg of a plasmid containing SARS-CoV-2 (2019-nCoV) Papain-like proteinase PLpro/NSP3 Gene ORF cDNA clone expression plasmid (Catalog Number: VG40593-UT; Sino Biological, Eschborn, Germany) or with pCMV3-untagged Negative Control Vector (Catalog Number: CV011; Sino Biological, Eschborn, Germany). The day after transfection, cells were cultured for 24 h in a medium containing HXT (Catalog No. S3826; Selleck Chemicals, Houston, TX, USA) at a concentration of 10 μM. All transfection experiments were performed with 80% confluent cells.

### 2.3. Quantitative Real-Time PCR (qRT-PCR)

Total RNA was extracted using the Total RNA Purification Plus Kit (Cat. No. 48300; Norgen, Thorold, Canada) according to the manufacturer’s protocol. Reverse transcription was performed using SuperScript VILO cDNA Synthesis (Ref: 11754-050, Thermo Fisher Scientific-Invitrogen). Gene expression analyses were performed by SYBR Green qRT-PCR. Based on the ΔΔCt method, relative amounts of mRNA were expressed as fold changes vs. control. Primer sequences, designed using Primers3Plus software and purchased by Merck-Sigma-Aldrich (Darmstadt, Germany) for measuring the expression of SARS-CoV-2 PLpro/NSP3, interferon α-1 (IFNA1), interferon α-7 (IFNA7), activating transcription factor 4 (ATF4), cyclic AMP-responsive element-binding protein 3 (CREB3), cyclic AMP-responsive element-binding protein 3-like protein 4 (CREB3L4), histone cluster 3, H3 (HIST3H3), histone cluster 1, H3f (HIST1H3F), histone cluster 1, H3h (HIST1H3H), MCF.2 cell line derived transforming sequence like (MCF2L), IL-1β, IL-6, TNF-α and Glyceraldehyde3-Phosphate Dehydrogenase (GAPDH), were reported in Appendix A. All SybrGreen reactions were carried out in 96-Well Reaction Plate (Cat. ECPCR0910C, Euroclone, Wotton, UK) in a total volume of 10 µL containing: 25 ng cDNA, forward and reverse primers, 2X SYBR Green PCR Master Mix (Cat. 44309155 Applied Biosystems, Carlsbad, CA, USA) and nuclease-free water. Reaction plates were cycled on a QuantStudio 7 Pro PCR System (Thermo Fisher Scientific-Applied Biosystems, Waltham, MA, USA) under the following conditions: 2 min at 50 °C, 10 min at 95 °C, then 40 cycles of 95 °C for 15 s and 60 °C for 1 min. GAPDH housekeeping gene was used as a reference control for normalization.

### 2.4. Total Protein Extraction and JESS Capillary Western Blotting

Total protein extraction was performed by homogenizing cells in Ripa lysis buffer (Merck-Sigma-Aldrich) and Halt Protease and Phosphate Inhibitor Cocktail (100X) (Thermo Fisher Scientific-Thermo Scientific), and incubated on ice for 30 min. The homogenates were then centrifuged at 13,000 rpm for 10 min and the resulting supernatant was taken as a protein sample and quantified using the BCA™ Protein Assay (Thermo Fisher Scientific-Thermo Scientific). Capillary western analyses were performed using the Jess Simple Western System (Bio-Techne, Minneapolis, MN, USA). Samples were diluted with 0.1X Sample Buffer. Then four parts of the diluted sample were combined with 1 part 5× Fluorescent Master Mix (containing 10× sample buffer and 400 mM DTT) and heated at 95 °C for 5 min. After this denaturation step, prepared samples, antibody diluent, primary antibodies, streptavidin-HRP, secondary conjugate, luminol-peroxide mix and wash buffer were dispensed into designated wells in an assay plate. A biotinylated ladder provided molecular weight standards for each assay. After plate loading, the separation electrophoresis and immunodetection steps took place in the fully automated capillary system. The following antibodies were used: (i) rabbit anti-SARS NSP3 (dilution 1:1000; Code: ab181620 Abcam, Cambridge, MA, USA); (ii) mouse anti-αTubulin (dilution 1:5000; code: nb100-690; Novus Biologicals, Littleton, CO, USA).

### 2.5. Human IFN Pathway

96-well plates are pre-configured with the most appropriate TaqMan^®^ Gene Expression Assays for a specific human IFN pathway (TaqMan Array96 well Human Interferon Pathway, Number Catalog 4414285; Thermo Fisher Scientific-Thermo Fisher). The contains 93 assays associated with interferon pathway genes and 3 assays to candidate endogenous control genes. 

Pathway analysis was conducted by querying Reactome annotations using the R/Bioconductor library reactome.db (accessed on 20 July 2022) [22]. For Reactome analysis, only pathways with an FDR lower than 0.05 and *p* < 0.05 were considered.

### 2.6. Cell Viability Assays 

Cell viability was evaluated using the Cell Proliferation Kit II-XTT (Merck-Roche Molecular Biochemicals, Basel, Switzerland), according to the manufacturer’s protocol. Two independent experiments were conducted. Into each well of a 96-well cell culture plate, 20,000 Calu-3, Caco-2 and HepG2 were seeded, and 24 h after plating were treated with or without HXT at different concentrations (5 μM, 10 μM, 20 μM and 30 μM). Assays were performed in quintuple. At the end of the treatment, 50 μL of a mixture of XTT labeling reagent and electron-coupling reagent were added and incubated at +37 °C for 4 h. The absorbance of the water-soluble formazan formed was measured at 492 and 620 nm using an ELISA microplate spectrophotometer.

### 2.7. Apoptosis

Apoptosis was assessed by Fluorescein-5-isothiocyanate (FITC) Annexin V Apoptosis Detection Kit I (code: 556547; Becton Dickinson-BD, Franklin Lakes, NJ, USA). Cells were seeded at a density of 100,000 cells/well in a 12-wells plate and were transfected with 800 nanograms of a plasmid containing SARS-CoV-2 (2019-nCoV) Papain-like proteinase/NSP3 Gene ORF cDNA clone expression plasmid or with pCMV3-untagged Negative Control Vector (all by Sino Biological Inc., Beijing, China). Briefly, cells were washed twice with cold PBS and resuspended in 1X Annexin Binding Buffer. Cells were then stained with 5 μL of FITC Annexin V and with 5 μM of Propidium Iodide (PI) for 15 min before analyzing. Acquisition and analysis were carried out on a FACSCanto II flow cytometer, using DiVa Software, version 6.3 (Becton, Dickinson and Company, Franklin Lakes, NJ, USA).

### 2.8. SDS-PAGE and Western Blotting

Total protein extraction was performed by homogenizing cells in Ripa lysis buffer (Merck-Sigma-Aldrich) and Halt Protease and Phosphatase Inhibitor Cocktail (100×) (Thermo Fisher Scientific-Thermo Scientific), and incubated on ice for 30 min. The homogenates were then centrifuged at 13,000 rpm for 15 min and the resulting supernatant was taken as a protein sample and quantified using the BCA™ Protein Assay (Thermo Fisher Scientific-Thermo Scientific). The samples were then diluted in the sample buffer 4× and resolved in 12% SDS-PAGE, then transferred and immobilized onto the nitrocellulose membranes (GE Healthcare, Munich, Germany). The membranes were blocked using 5% non-fat dry milk for 30 min and incubated with the appropriate primary and secondary antibodies. Protein expression was quantified by densitometric analysis using the open source Image J v3.91 software (https://imagej.nih.gov/ij/download.html, accessed on 20 July 2022). As antibodies were used: (i) rabbit anti- active caspase-8 antibody (dilution 1:1000; code: NB100-56116; Novus Biological LLC, Centennial, CO, USA); (ii) rabbit anti-GAPDH (dilution 1:1000; code 5174; Cell Signaling, Danvers, MA, USA). 

### 2.9. Inflammation Cytokine Array

One-hundred micrograms of proteins were used for the analysis of 40 cytokines (Eotaxin, Eotaxin-2, GCSF, GM-CSF, ICAM-1, IFN-gamma, I-309, IL-1α, IL-1β, IL-2, IL-3, IL-4, IL-6, IL-6sR, IL-7, IL-8, IL-10, IL-11, IL-12p40, IL-12p70, IL-13, IL-15, IL-16, IL-17, IP-10, MCP-1, MCP-2, M-CSF, MIG, MIP-1α, MIP-1β, MIP-1delta, RANTES, TGF-β1, TNF-α, TNF-β, sTNF RI, sTNF-RII, PDGF-BB, TIMP-2) using the Human Inflammation Antibody Array-Membrane Kit (ab134003; Abcam, Cambridge, UK) following the manufacturer’s instructions. In brief, the array membranes were blocked by incubation with a blocking buffer for 30 min. After incubation, membranes were incubated with biotin-conjugated anti-cytokines overnight at +4 °C and then with HRP-Conjugated Streptavidin for 2 h. The signal was detected by chemiluminescence with iBright Western Blot Imaging Systems (Thermo Fisher Scientific-Thermo Fisher). Quantification of each protein spot was performed by measuring integrated density using densitometry Image J v3.91 software.

### 2.10. HPLC to Analyze Various Forms of Glutathione

Reduced and oxidized glutathione (GSH and GSSG, respectively) were analyzed by HPLC as previously described [23]. In detail, for glutathione forms determination, cells were mixed with 100 μL of 10 mmol/L phosphate buffer, pH 7.2 (for GSH), or with 100 μL of the same buffer containing 5 mmol/L N-ethylmaleimide (for GSSG, and protein-bound glutathione, GS-Pro). Cells are then lysed by sonication three times for 2 s. After sonication, 50 μL of 12% sulfosalicylic acid was added; the protein pellet is dissolved in 150 μL of 0.1 M NaOH, and the glutathione content in the acid-soluble fraction is determined. Proteins were measured using BCA™ Protein Assay (Thermo Scientific, Rockford, IL, USA). For GS-Pro determinations, the protein pellet was re-suspended in 150 μL of NaOH 0.1 M and derivatized. The levels of the different forms of glutathione were determined by using the derivatization and chromatography procedures. Total glutathione (TotGSH) amounts were calculated by the sum of free GSH, GSSG and GS-Pro. GSH was calculated by subtracting GSSG to free GSH.

### 2.11. Determination of the Oxidative Stress Parameters

Determination of the oxidative stress parameters was carried out on 2 × 10^7^ cells for each treatment collected by centrifugation at 1500× *g* for 10 min at 4 °C. Cell lysates were obtained by treating the cell pellets with 1 mL of cold phosphate buffer saline (PBS, pH 6.7) containing 1 mM EDTA, sonicating in an ice-cold bath for 5 min using a 3 mm titanium probe set to 200 W and 30% amplitude (Vibra Cell™ Sonics Materials, Inc., Danbury, CT, USA). PBS was used as a blank in each assay. The intracellular Reactive Oxygen Species (ROS) levels, expressed as percentage (%), were assessed according to Smeriglio et al. [24] by recording the fluorescence resulting from the intracellular oxidation of 2′,7′-dichlorofluorescin diacetate (DCF-DA). The probe, diluted in PBS (10 μM) was added, and the cell was incubated for 30 min to label intracellular ROS. The medium was then removed, and the cells were washed five times with 1× PBS. Fluorescence of the labeled intracellular ROS was recorded by a plate reader (FLUOstar Omega, BMG LABTECH, Ortenberg, Germany) at the following excitation and emission wavelengths: λ_ex_ 485 nm; λ_em_ 535 nm. The release of Nitric Oxide (NO) was evaluated according to Smeriglio et al. [24] by Griess’s reagent. The absorbance was measured at 550 nm using a UV-Vis plate reader (Multiskan GO, Thermo Scientific, Waltham, MA, USA). NO was quantified using sodium nitrite as a reference standard (1.0–15 µM). The measurement of Thiobarbituric Acid Reactive Substances (TBARS) was carried out by TBARS (TCA Method) assay kit Item No. 700870 (Cayman Chemical, Ann Arbor, MI, USA) following the manufacturer’s instructions. The pink malondialdehyde (MDA)-thiobarbituric acid (TBA) adduct, formed by the reaction of MDA and TBA under high temperature (90–100 °C) and acidic conditions, was measured colorimetrically at 540 nm by using the UV-Vis plate reader reported above, and using MDA as the reference standard (0.625–50 µM).

The protein carbonyl colorimetric assay kit Item No. 10005020 (Cayman Chemical, Ann Arbor, MI, USA) based on 2,4-dinitrophenylhydrazine reaction, was used to quantify the protein carbonyl content according to the manufacturer’s instructions. The amount of protein-hydrazone produced (nM) was quantified spectrophotometrically at 370 nm by using the UV-Vis plate reader reported above. Both kits provide a simple, reproducible, standardized and validated (precision, intra-assay and inter-assay coefficient of variation, and recovery) tool for assaying lipid peroxidation and protein carbonyl content in cell lysate, respectively. Furthermore, several reagents were tested by the manufacturer for interference such as buffers, detergents, protease inhibitors and chelators, highlighting that no interferences can occur in the experimental condition adopted in the present study. All data were normalized for protein concentration.

### 2.12. Statistical Analysis

The data are presented as mean ± standard deviation (SD). Comparisons were made between means from at least two independent experiments repeated in duplicate. Statistical differences were analyzed using the Student’s *t*-test. Values of *p* < 0.05 were considered to be statistically significant. Data analysis was performed with GraphPad Prism 9.0 (GraphPad Software, San Diego, CA, USA).

## 3. Results

### 3.1. Expression of SARS-CoV-2-PLpro Induced an Up-Regulation of Genes of the IFN Pathway and of Pro-Inflammatory Cytokines

Polarized human Calu-3, Caco-2, and HepG2 epithelial cells were transfected with the SARS-CoV-2-PLpro (PLpro) plasmid or with an empty vector (pCMV3) as confirmed by the PLpro gene (Figure 1A) and protein (Appendix A) expression. Since PLpro plays a key role in triggering IFN-related signaling and pro-inflammatory molecules in host cells, the expression of 93 genes related to the IFN pathway was evaluated in all epithelial cell lines. Gene array analysis was performed in Calu-3 cells expressing PLpro, compared to cells expressing pCMV3 (Appendix A). As shown in Appendix A, results in PLpro Calu-3 cells showed an up-regulation of nine interferon-related genes, including IFN1A, IFNA7, ATF4, CREB3, CREB3L4, HIST1H3F, HIST1H3H, HIST3H3 and MCF2L, respect to pCMV3; whereas no IFN-related genes were significantly down-regulated. Reactome enrichment analysis for the up-regulated genes showed that these genes belong to the following top-ten pathways: Cellular responses to stress; Cellular responses to stimuli; Factors involved in megakaryocyte development and platelet production; Hemostasis; TRAF6-mediated IRF7 activation; DDX58/IFIH1-mediated induction of interferon-α/β; Unfolded Protein Response; Cytokine Signaling in Immune system; Innate Immune System; and Infectious disease (Figure 1B). These results were confirmed by a single SYBR Green qRT-PCR assay in Calu-3, Caco-2, and HepG2 epithelial cells (Figure 1C–E). Moreover, the exogenous expression of PLpro also caused the transcriptional increase in genes encoding for pro-inflammatory cytokines such as IL-1β, IL-6 and TNF-α (Figure 1F–H).

### 3.2. The Treatment with HXT Reduced the PLpro mRNA Expression and the Apoptosis

HXT, one of the main phenolic compounds in olive oil, has several biological activities, including anti-inflammatory, anti-microbial and antioxidant properties [20].

In order to evaluate if the treatment with HXT was able to reduce the effects of exogenous expression of PLpro, Calu-3, Caco-2 and HepG2 cells were treated with different concentrations of HXT (5 μM, 10 μM, 20 μM and 30 μM). Cell viability was unaffected by PLpro expression in all cell lines, while it already significantly increased after the treatment with 5 μM HXT for 24 h and reached a plateau at 10 μM HXT (Appendix A). Therefore, this latter concentration of HXT was chosen for the next experiments.

As shown in Figure 2A, the exposure of Calu-3, Caco-2 and HepG2 cells to 10 μM HXT (PLpro + HXT) caused a statistically significant reduction of PLpro mRNA with respect to untreated counterpart (PLpro) after 24 h.

A recent study reported that SARS-CoV-2 infection could activate caspase-8 to induce apoptosis [25]. Therefore, we evaluated apoptosis by Annexin V assay and the expression of active caspase-8. Exogenous expression of PLpro increased the apoptotic rate in all cell lines compared to cells with empty vector, whereas the treatment with 10 μM HXT reduced apoptosis when compared to their untreated counterpart (Figure 2B and Appendix A).

### 3.3. The Treatment with HXT Reduced PLpro-Dependent Inflammatory Response

Next, we tested the ability of HXT into reducing the PLpro-dependent inflammatory response in polarized human epithelial cells. The inflammatory profile was evaluated in terms of both gene and protein expression levels. As shown in Figure 3A–C, the exposure to 10 μM HXT caused a statistically significant reduction of mRNA levels of several IFN-related genes and cytokines in the treated cells (PLpro + HXT) compared to untreated cells expressing viral peptide (PLpro). In particular, the data highlights that HXT significantly down-regulated: IFN7A, IL-1β, IL-6 and TNF-α gene expression in all cell lines; IFN1A and CREB3 gene expression in Calu-3 and Caco-2; CREB3L4 and MCF2L gene transcription in Caco-2 and HepG2; HIST1H3H gene expression levels in Calu-3 and HepG2; HIST3H3 and HIST1H3F gene transcription in Calu-3 only; and ATF4 gene expression in Caco-2 only (Figure 3A–C).

Moreover, the evaluation of the effects of PLpro on the inflammatory pattern was also performed by the analysis of a panel of 40 anti/pro-inflammatory cytokines by Human inflammation array (Appendix A). As reported in Figure 4A–C and Appendix A exogenous expression of PLpro caused changes in the protein expression levels of several cytokines and chemokines. Among the significantly up-regulated and down-regulated pro-inflammatory molecules, seven were common to all polarized human epithelial cells (Figure 4D,E).

Finally, as shown in Figure 5A-C, the treatment with 10 μM HXT significantly reduced the expression levels of EOTAXIN, EOTAXIN-2, IL-1β, IL-2, IL-6, and TNF-α in all cell lines, while the natural antioxidant significantly decreased the expression levels of IL-3 in Caco-2 cells only. These effects of HTX were not ascribable to a direct anti-inflammatory action on cells but rather to an indirect action on PLpro expression (Appendix A).

### 3.4. Expression of SARS-CoV-2-PLpro Induced Oxidative Stress That Was Counteracted by the Treatment with HXT

Unbalanced redox homeostasis and accumulation of ROS seem to be crucial to promoting most of the inflammatory conditions linked to COVID-19 [26,27]. These effects have been associated with the activity of specific viral proteins, such as the binding of SARS-CoV-2-Spike protein to ACE-2, but to date, the possible contribution of PLpro remains to be explored. Therefore, here we analyzed the effect of PLpro in polarized human epithelial cells, and then we evaluated whether HXT was able to counteract these effects.

As shown in Figure 6A–E, the exogenous expression of PLpro caused a significant decrease in GSH/GSSG ratio, a significant increase in ROS release percentage, NO levels, TBARS levels and protein carbonyl levels in Calu-3, Caco-2 and HepG2 cells.

Finally, the treatment of PLpro expressing cells with 10 μM HXT for 24 h caused a significant increase in GSH/GSSG ratio, and a significant decrease in all oxidative stress markers (Figure 7A–E).

## 4. Discussion

Here, we reported for the first time the effects of the exogenous expression of the catalytic subunit PLpro SARS-CoV-2 NSP3 protein in the polarized human airway, intestinal, and liver epithelial cells. Our results demonstrated that PLpro was able to induce a cascade of inflammatory genes and proteins, apoptosis and a strong increase in several oxidative stress markers. Of note, we demonstrated that HXT reversed the effects on most of these altered pathways.

SARS-CoV-2 infection was initially described as causing severe respiratory disease, but several clinical and experimental studies have demonstrated that infected individuals may exhibit a severe inflammatory response, which in turn results in a severe multi-organ dysfunction and finally death [28]. In addition, even patients with mild or asymptomatic disease may present a variety of symptoms including fatigue, intermittent fever, shortness of breath, cough, joint, chest and muscle pain. These clinical manifestations persisting for more than 12 weeks after acute infection have been called long-COVID syndrome, named also PASC (post-acute sequelae of SARS-CoV-2). Patients with long-COVID are often defined as long haulers [12]. The clinical manifestations may derive from the involvement of tissues and cells for which SARS-CoV-2 has an extended tropism, including lung, intestinal and liver epithelia and could be linked to the interplay between chronic inflammation and persistence of viral proteins [8,17,29]. Regarding mechanisms explaining long-COVID syndrome, there are various hypotheses, such as the presence of persistent viral reservoirs in the body and harmful immune response. Recently, SARS-CoV-2 RNA persistence has been reported in monocytes and the gut [30,31]. Even though the persistence NSPs has not yet been reported, NSP3, NSP4 and NSP6, may contribute to forming double-membrane vesicles where these proteins could be retained and hidden to host cell defense [32]. Therefore, we hypothesize that SARS-CoV-2 PLpro/NSP3 of SARS-CoV-2 could persist for several months after acute infection, thus playing a central role in the pathogenesis of long-COVID syndrome. In this scenario, the catalytically active PLpro may play a dual role in COVID-19. On the one hand, PLpro may enzymatically reduce the ISGylation of MDA5 and IRF3 genes, thus inhibiting the host response to viral infection [33]; on the other PLpro may induce the extracellular release of ISG15, thus amplifying the production and release of pro-inflammatory cytokines and chemokines in host infected cells [34]. A similar phenomenon could be caused by the presence of virally inert SARS-CoV-2 RNA and peptides that could remain in host cells also after virus eradication and trigger the onset of long-COVID syndrome [35]. However, further studies are required to support our hypothesis.

Here, we show that polarized human airway, intestinal and liver epithelial cells expressing exogenous PLpro up-regulate the expression of IFN-related genes (i.e., IFNA1, IFNA7, ATF4, CREB3, CREB3L4, HIST3H3, HIST1H3F, HIST1H3H, MCF2L) and pro-inflammatory cytokine genes/proteins (mainly IL-1β, IL-6 and TNF-α). These results are in line with previous studies that highlighted the role of PLpro expression in human cells and its ability to establish a functional interactome with host factors that elicit anti-viral signaling and inflammatory response [33,36,37]. The activation of inflammatory signals in response to SARS-CoV-2 infection response may induce five kinds of regulated cell death, including apoptosis, necroptosis, pyroptosis, autophagy and PANoptosis. In the proposed model, SARS-CoV-2-induced apoptosis was mediated by caspase-8 activation and mitochondrial pathways [38]. SARS-CoV-2 ORF3 has been reported as a pivotal inductor of apoptosis in several cell lines, even though more moderately than its SARS-CoV homolog [39]. Accordingly, we demonstrated for the first time that SARS-CoV-2 PLpro/NSP3 induced increased expression of active caspase-8 and consequent apoptosis. Evidence in cells expressing SARS-CoV 3CLpro reported a significant increase in apoptotic rate associated with ROS production [40]. This phenomenon was also confirmed in our polarized human epithelial cells, where exogenous expression of SARS-CoV-2 PLpro/NSP3 caused a decrease in GSH/GSSG and induction of several markers of oxidative stress (i.e., ROS, NO, TBARS and protein carbonyl).

Our data and other reported evidence suggest that SARS-CoV-2 PLpro/NSP3 may be a highly attractive and druggable target not only in acute COVID-19 but also in long-COVID [41,42]. Moreover, several lines of evidence demonstrated that inflammation and oxidative stress mutually reinforce each other in COVID-19 and presumably also in long-COVID, highlighting the major role of oxidative unbalance as a trigger of both acute and chronic inflammation [27,43]. Therefore, it is plausible that natural antioxidant molecules may counteract SARS-CoV-2 redox status derangement and consequent inflammation representing a possible therapy for improving signs and symptoms of long-COVID syndrome [26,27,44]. Accordingly, recently, Bartolini et al. demonstrated that SARS-CoV-2 infection may cause a marked decrease in cellular thiols, mainly GSH, and that the antioxidant N-acetyl-cysteine may cooperate with Nelfinavir restoring protective levels of GSH in VERO E6 cells [45].

HXT is a phenolic compound found in the leaves and fruits of olive with antioxidant, anti-inflammatory and antimicrobial activities [20,21]. The antiviral activity of olive leaf metabolites against SARS-CoV-2 was reported in several in silico computational studies [43,44]. In particular, Yu et al. tested several viral targets, such as viral proteases (Mpro/3CLpro, PLpro), TLRs, ACE2, RBD, NSP15, HSPA5 SBDβ, TMPRSS2, S protein and furin [46]. Furthermore, Takeda et al. showed the antiviral activities of HXT against SARS-CoV-2. In particular, an HXT-rich cream showed virucidal activity through the induction of structural changes in SARS-CoV-2, by modifying the molecular weight of the spike proteins and disrupting the viral genome [47]. In addition, HXT attenuated the pro-inflammatory agents in both in vitro and in vivo disease models [48,49,50,51].

Hence, here we investigated whether the treatment with HXT may improve inflammation, apoptosis and oxidative stress induced by SARS-CoV-2 PLpro/NSP3 in the polarized human airway, intestinal and liver epithelial cells. Confirming our hypothesis, our results demonstrated that the treatment of PLpro expressing cells with HXT restored the expression of pro-inflammatory genes/proteins at levels similar to those expressed in controls, reduced apoptotic rate and pro-oxidant state. It is plausible that the HXT-dependent reduction of inflammation could be mainly due to its capacity in reducing the expression of PLpro in infected cells. These findings increase the chances that HXT could be used as a preventive treatment to avoid long-COVID development.

## 5. Conclusions

Many researchers have focused on designing drugs, which can affect the replication or protease activity of SARS-CoV-2 to reduce severe/mild COVID-19 disease, but currently, there is an urgent need for the requirement of safe and effective treatments to alleviate long-COVID syndrome. Natural antioxidants, such as HXT, may have health benefits in this scenario.

Here, we provide the first evidence that SARS-CoV-2 PLpro promotes pro-inflammatory, pro-apoptotic and pro-oxidants effects in epithelial cells, thus suggesting that if this portion of NSP3 remains in host cells after viral clearance it could participate in long-COVID pathogenesis. Further clinical and experimental studies are needed to confirm this hypothesis. Moreover, we demonstrate that PLpro-dependent adverse effects are reversed by the treatment with HXT, thus indicating the possible effectiveness of this molecule as a longer-term and safe approach to reduce the symptoms of long-COVID, in adults, but also in children and adolescents where this condition is a relevant, unrecognized health problem [52].

## Figures and Tables

**Figure 1 antioxidants-11-01466-f001:**
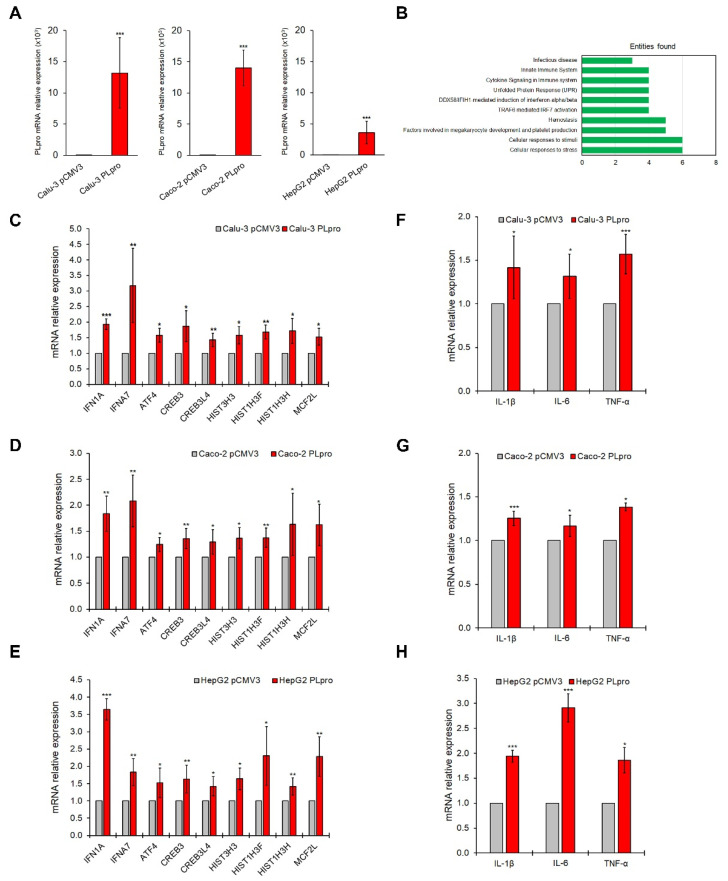
PLpro expression up-regulates genes of IFN pathways and pro-inflammatory cytokines in epithelial cells. (**A**) PLpro mRNA expression in Calu-3, Caco-2 and HepG2 cells after transfection with plasmid for PLpro (PLpro) or empty vector (pCMV3). (**B**) Bar plots of the 10 most abundant pathways for commonly up-regulated genes investigated in Calu-3, Caco-2 and HepG2 cells after transfection with plasmid for PLpro (PLpro) or empty vector (pCMV3). Relative mRNA expression of IFNA1, IFNA7, ATF4, CREB3, CREB3L4, HIST3H3, HIST1H3F, HIST1H3H, MCF2L genes measured by qRT-PCR in Calu-3 (**C**), Caco-2 (**D**), and HepG2 (**E**) cells transfected with plasmid for PLpro (PLpro) or empty vector (pCMV3). Relative mRNA expression of IL-1β, IL-6 and TNF-α genes measured by qRT-PCR in Calu-3 (**F**), Caco-2 (**G**), and HepG2 (**H**) cells transfected with plasmid for PLpro (PLpro) or empty vector (pCMV3). Data are the mean ± SD of three independent experiments repeated at least in triplicate. Data were analyzed by 2-tailed *t*-tests, * *p* < 0.05, ** *p* < 0.01, *** *p* < 0.001.

**Figure 2 antioxidants-11-01466-f002:**
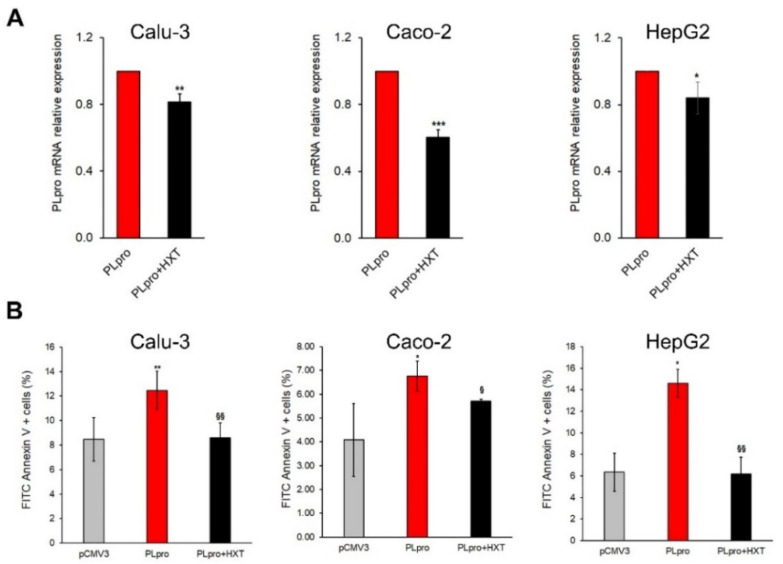
Treatment with HXT down-regulates PLpro mRNA expression and reduces apoptosis in epithelial cells. (**A**) PLpro mRNA expression in Calu-3, Caco-2 and HepG2 cells before and after treatment with 10 μM HXT for 24 h. (**B**) Percentage of apoptosis measured by FITC Annexin V in pCMV3 Calu-3, Caco-2 and HepG2 cells, and in PLpro cells treated or not with 10 µM HXT for 24 h. Values are plotted as mean ± SD of three independent experiments repeated at least in duplicate. Data were analyzed by 2-tailed *t*-tests, * *p* < 0.05, ** *p* < 0.01, *** *p* < 0.001 vs. pCMV3; ^§^ *p* < 0.05, ^§§^ *p* < 0.01 vs. PLpro.

**Figure 3 antioxidants-11-01466-f003:**
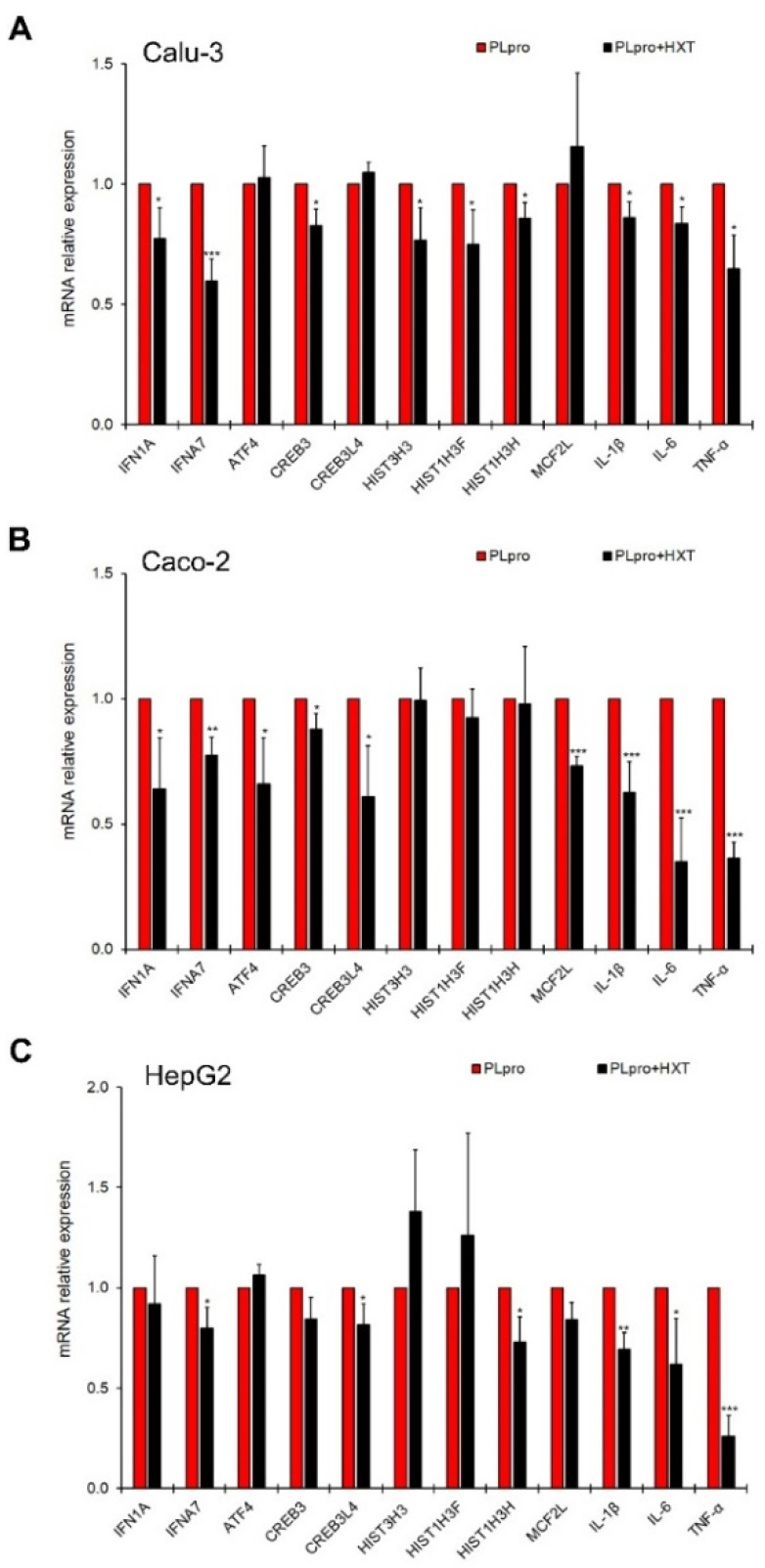
Treatment with HXT down-regulates pro-inflammatory genes in epithelial cells. Relative mRNA expression of IFNA1, IFNA7, ATF4, CREB3, CREB3L4, HIST3H3, HIST1H3F, HIST1H3H, MCF2L, IL-1β, IL-6 and TNF-α genes measured by qRT-PCR in Calu-3 (**A**), Caco-2 (**B**), and HepG2 (**C**) cells expressing PLpro before and after treatment with 10 μM HXT for 24 h. Values represented mean ± SD of two independent experiments repeated at least in duplicate. Data were analyzed by 2-tailed *t*-tests, * *p* < 0.05, ** *p* < 0.01, *** *p* < 0.001 vs. PLpro.

**Figure 4 antioxidants-11-01466-f004:**
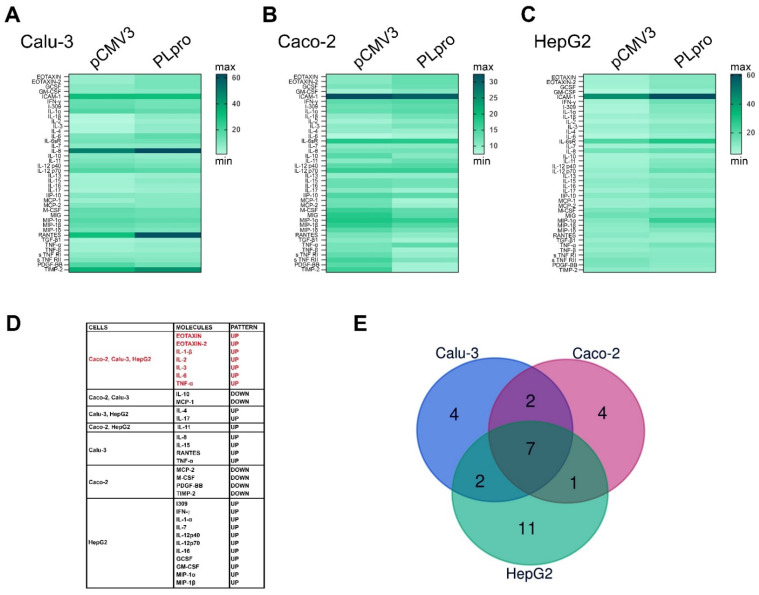
PLpro alters the expression of pro-inflammatory proteins in epithelial cells. Heatmap representation of semi-quantitative expression of 40 pro-inflammatory cytokines and chemokines by antibody array in Calu-3 (**A**), Caco-2 (**B**), and HepG2 cells (**C**) expressing PLpro compared to their pCMV3 counterpart. (**D**) Box reporting the list of significantly up- or down-regulated pro-inflammatory proteins in all cell lines after transfection with PLpro (PLpro) compared to empty vector (pCMV3). (**E**) Venn diagram of commonly altered pro-inflammatory molecules in Calu-3, Caco-2 and HepG2 expressing PLpro. Data are reported as mean of Integrated Density values of two independent experiments repeated at least in duplicate.

**Figure 5 antioxidants-11-01466-f005:**
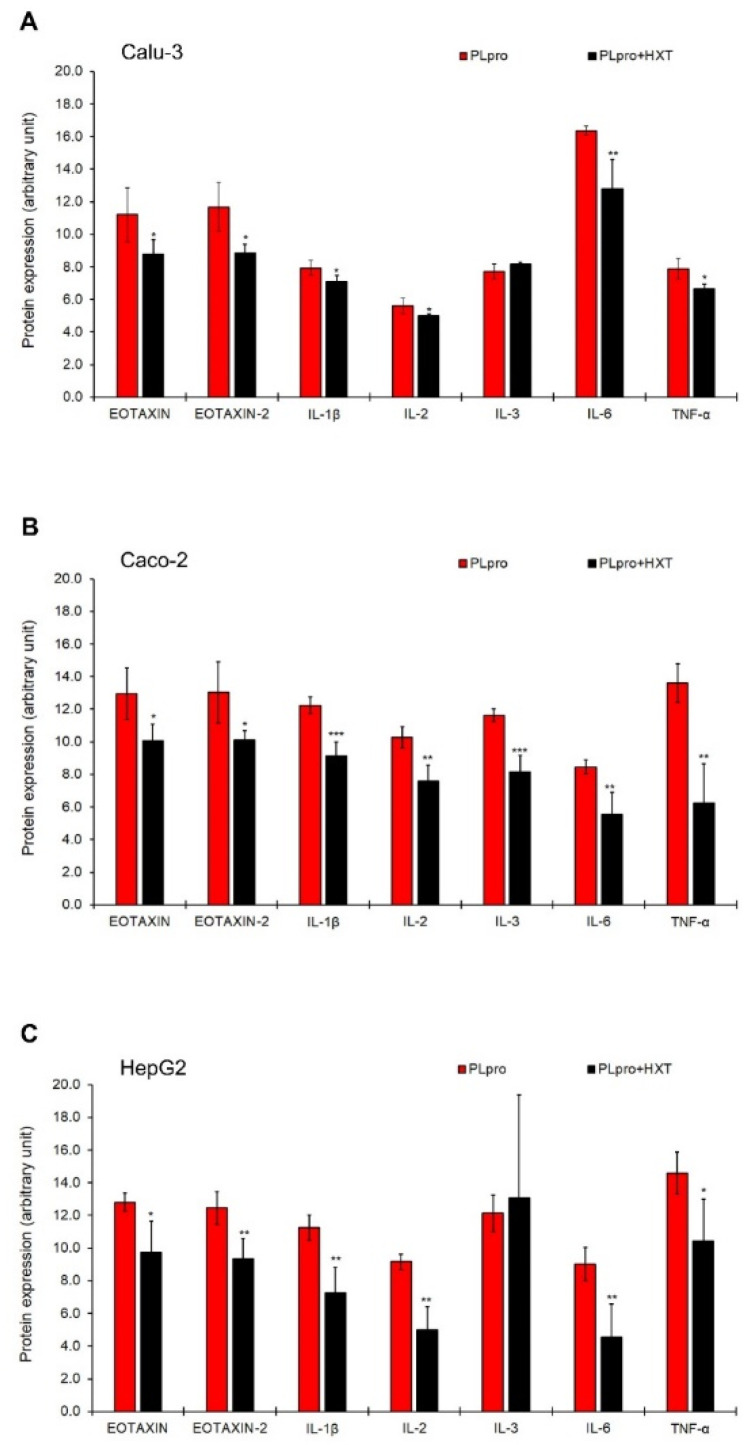
Treatment with HXT down-regulates pro-inflammatory proteins induced by PLpro expression in epithelial cells. Semi-quantitative expression of EOTAXIN, EOTAXIN-2, IL-1β, IL-2, IL-3, IL-6, and TNF-α in Calu-3 (**A**), Caco-2 (**B**), and HepG2 (**C**) cells expressing PLpro, before and after treatment with 10 μM HXT for 24 h. Values are plotted as mean ± SD of two independent experiments repeated in duplicate. Data were analyzed by 2-tailed *t*-tests, * *p* < 0.05, ** *p* < 0.01, *** *p* < 0.001 vs. PLpro.

**Figure 6 antioxidants-11-01466-f006:**
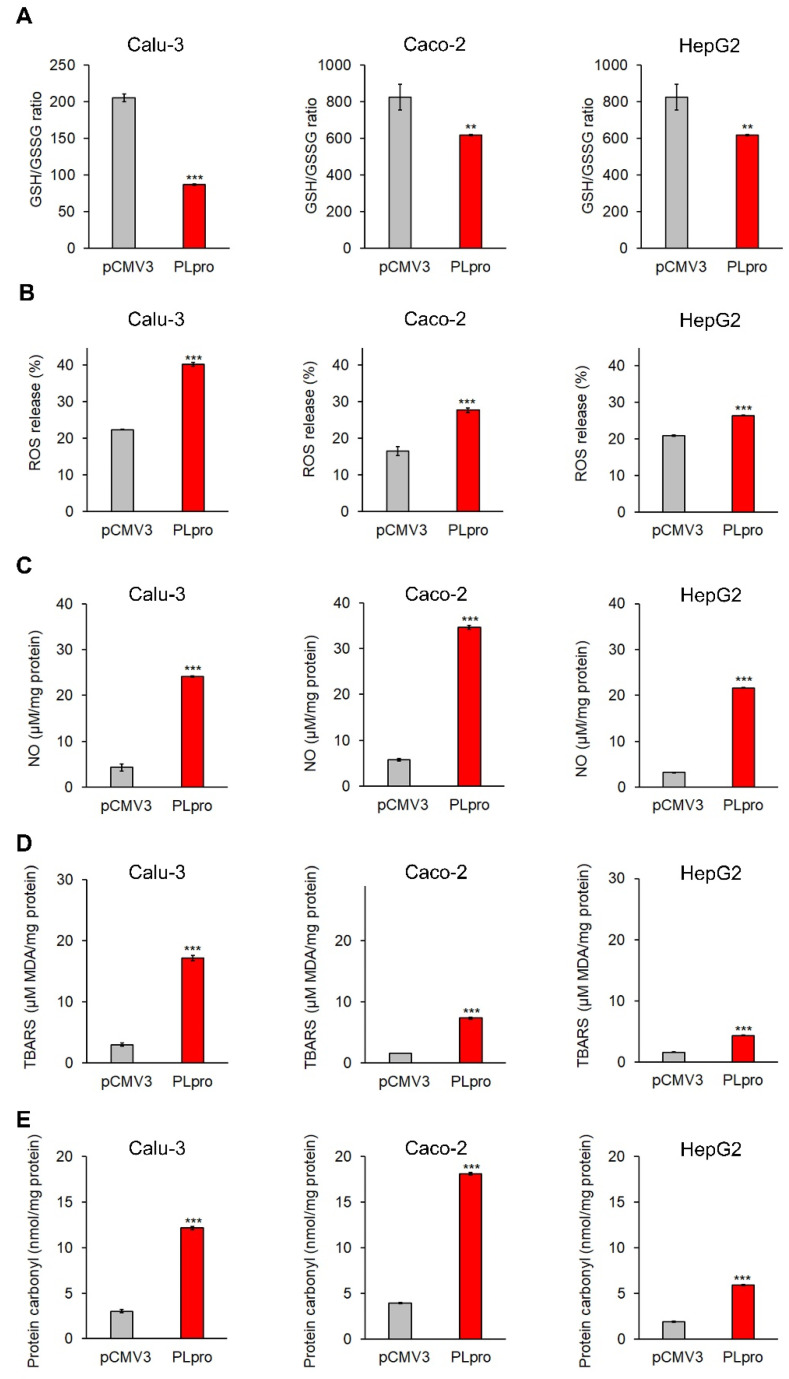
PLpro expression causes pro-oxidant response in epithelial cells. The histograms report GSH/GSSG ratio (**A**), percentage of ROS release (**B**), and levels of NO (**C**), TBARS (**D**) and protein carbonyl (**E**) in Calu-3, Caco-2 and HepG2 cells expressing PLpro (PLpro) compared to empty vector (pCMV3). Values are plotted as mean ± SD of two independent experiments repeated in duplicate. Data were analyzed by 2-tailed *t*-tests, ** *p* < 0.01, *** *p* < 0.001 vs. pCMV3.

**Figure 7 antioxidants-11-01466-f007:**
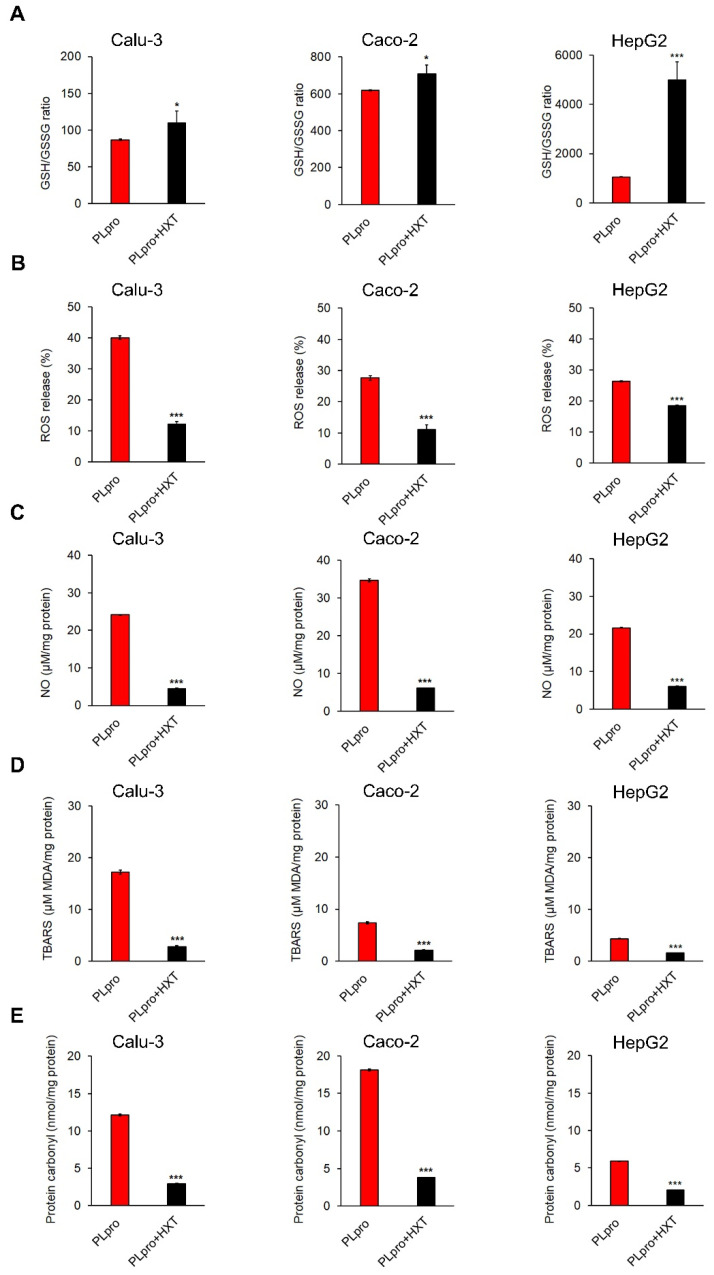
Treatment with HXT reduces PLpro-dependent pro-oxidant response in epithelial cells. The histograms report GSH/GSSG ratio (**A**), percentage of ROS release (**B**), and levels of NO (**C**), TBARS (**D**) and protein carbonyl (**E**) in Calu-3, Caco-2 and HepG2 cells expressing PLpro, before and after treatment with 10 μM HXT for 24 h. Values are plotted as mean ± SD of two independent experiments repeated in duplicate. Data were analyzed by 2-tailed *t*-tests, * *p* < 0.05, *** *p* < 0.001 vs. PLpro.

## Data Availability

Data are contained within the article and Appendix A.

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
