# Peer review of "Hydroxytyrosol Recovers SARS-CoV-2-PLpro-Dependent Impairment of Interferon Related Genes in Polarized Human Airway, Intestinal and Liver Epithelial Cells"

_antioxidants, 2022, doi:10.3390/antiox11081466_

Round 1

Reviewer 1 Report

Dear Authors,

Overall, this manuscript is well-written and it addresses a topic of great importance. The authors describe an interesting possible mechanism that SARS-CoV-2 PL promotes pro-inflammatory, pro-oxidants, and pro-apoptotic activities in human epithelial cells and that these effects are reversed by the treatment with HTS.

Issues that the author should consider in order to strengthen the manuscript are as follows:

(I)              Please select your words carefully so that the Abstract (background) and Introduction do not get too long.

(II)             Please include sufficient detail in Methods section so that experimental procedures can be repeated by other researchers in the field according to MDPI recommendations.

The Authors investigated the protective effects of HTS on human epithelial cells against SARS-CoV-2 PLpro in vitro and different methods are used to evaluate biomarkers of oxidative/nitrosative stress: assessment of antioxidant status, direct measurement of reactive oxygen species, protein damage, and end-products of peroxidation of polyunsaturated fatty acids. The oxidative chemistry of such oxidative/nitrosative end-products must be considered when predicting how antioxidants could behave in vitro or in vivo. Cells in culture are exposed to redox-active transitions metals, oxygen (O2), and added antioxidants to growth culture media may generate H2O2 and change cell behavior.

Please report buffers, incubation media and mycoplasma contamination status of the cell lines used.

1.      HPLC to analyze various forms of glutathione

2.      Determination of the oxidative stress parameters: (a)The intracellular Reactive Oxygen Species(ROS); (b) The measurement of Thiobarbituric Acid Reactive Substances (TBARS); The pink malonyl-dialdeide (MDA)-thiobarbituric acid (TBA) adduct; (c)The protein carbonyl colorimetric assay

The measurement of redox-related biomarkers gives insights into how oxidative damage is modulated in vivo in human disease. However, the measurement of these biomarkers requires validated methods, mostly based on quantitative MAS (mass spectrometry) and researchers should be wary of “kit-based” methods characterized by a very low validity.

Cooke MS. A commentary on ‘urea, the most abundant component in urine, cross‐reacts with a commercial 8-OH-dG ELISA kit and contributes to overestimation of urinary 8-OH-dG’. What is ELISA detecting? Free Radic Biol Med. 2009;47:30–1.

Halliwell B. 5th. Oxford University Press; 2015. Free Radicals in Biology and Medicine edn. Oxford

(III)            Please highlight the limitations and implications for researchers carrying out future studies in the same field.

Below, please find other suggestions:

KEYWORDS: SARS-CoV-2; C-19???; Hydroxytyrosol; Cytokines; Interferon; Oxidative stress. (The title words should not be repeated in Keywords).

INTRODUCTION:

Line 69-70: Like As well as for other respiratory viruses, SARS-CoV-2 transmission occurs mainly through the respiratory route.

Line 83-84: Lines of evidence suggest that systemic and tissue chronic  chronic tissue inflammation, autoimmunity and viral protein/RNA persistence might contribute to the  pathogenesis of long-COVID  [15-17].

DISCUSSION:

Line 483-485: On the one hand, Plpro may enzymatically may reduce the ISGylation of MDA5 and IRF3 genes, thus inhibiting the host response to viral infection [30];  

Line 511-514: Moreover, several lines of evidence demonstrated that inflammation and oxidative stress are mutually reinforce each other in COVID-19 and presumably also in long-COVID, highlighting the major role of oxidative unbalance as a trigger of both acute and chronic inflammation[27,40].

Line 514-517: Therefore, it is plausible that natural antioxidant molecules may counteract SARS-CoV-2 redox status derangement and consequent inflammation, representing a possible therapy for improve improving signs and symptoms of long-COVID syndrome [26,27,41].

 CONCLUSIONS:

Line 540-542: Many researchers have focused on designing drugs, which can affect replication or protease activity of SARS-CoV-2 to reduce severe/mild COVID-19 disease but, currently, there is an urgent need for  requirement of  safe and effective treatments to alleviate long-COVID syndrome.

Line 548-556: Moreover, we demonstrate that PLpro-dependent adverse effects are reversed by the treatment with HXT, thus indicating the possible effectiveness of this molecule as a longer-term and safe approach to reduce the symptoms of long-COVID in adults, but also in children and adolescents where this condition is a relevant, unrecognized health problem [49].

Author Response

Reviewer 1

Dear Authors,

Overall, this manuscript is well-written and it addresses a topic of great importance. The authors describe an interesting possible mechanism that SARS-CoV-2 PL promotes pro-inflammatory, pro-oxidants, and pro-apoptotic activities in human epithelial cells and that these effects are reversed by the treatment with HTS.

Reply to general comment: We thank this Reviewer for appreciating our manuscript and for suggestions that we have follow to improve the manuscript. All changes/addition were highlighted in yellow.

Issues that the author should consider in order to strengthen the manuscript are as follows:

Q1: Please select your words carefully so that the Abstract (background) and Introduction do not get too long.

Reply to Q1: We thank the Reviewer 1 for his/her suggestions. We have revised the Background of Abstract and Introduction of our manuscript by reducing the text and adding corrections that have been requested by all Reviewers.

Q2: Please include sufficient detail in Methods section so that experimental procedures can be repeated by other researchers in the field according to MDPI recommendations.

The Authors investigated the protective effects of HTS on human epithelial cells against SARS-CoV-2 PLpro in vitro and different methods are used to evaluate biomarkers of oxidative/nitrosative stress: assessment of antioxidant status, direct measurement of reactive oxygen species, protein damage, and end-products of peroxidation of polyunsaturated fatty acids. The oxidative chemistry of such oxidative/nitrosative end-products must be considered when predicting how antioxidants could behave in vitro or in vivo. Cells in culture are exposed to redox-active transitions metals, oxygen (O2), and added antioxidants to growth culture media may generate H2O2 and change cell behavior.

Please report buffers, incubation media and mycoplasma contamination status of the cell lines used.

  1. HPLC to analyze various forms of glutathione
  2. Determination of the oxidative stress parameters: (a)The intracellular Reactive Oxygen Species(ROS); (b) The measurement of Thiobarbituric Acid Reactive Substances (TBARS); The pink malonyl-dialdeide (MDA)-thiobarbituric acid (TBA) adduct; (c)The protein carbonyl colorimetric assay The measurement of redox-related biomarkers gives insights into how oxidative damage is modulated in vivo in human disease. However, the measurement of these biomarkers requires validated methods, mostly based on quantitative MAS (mass spectrometry) and researchers should be wary of “kit-based” methods characterized by a very low validity.

Cooke MS. A commentary on ‘urea, the most abundant component in urine, cross‐reacts with a commercial 8-OH-dG ELISA kit and contributes to overestimation of urinary 8-OH-dG’. What is ELISA detecting? Free Radic Biol Med. 2009;47:30–1.

Halliwell B. 5th. Oxford University Press; 2015. Free Radicals in Biology and Medicine edn. Oxford

Reply to Q2. We thank the Reviewer for his careful evaluation of the present study and for his/her suggestions and comments. The materials and methods concerning the determination of the oxidative stress parameters was revised accordingly, including also details of sample preparation and assays performed. Moreover, more details about the kit used for TBARS and carbonyl proteins determination were added, highlighting the specificity (including interfering substances) and validation of the kits used for determination of these oxidative markers in cell lysates. All these specifications can be consulted online on the kit manufacturer's website. Finally, the authors are keen to underline that these kits are routinely used for the determination of these parameters in different types of biological samples as can be deduced from the following recent publications reported only by way of example:

  • Cong, C., Yuan, X., Hu, Y., et al. Sinigrin attenuates angiotensin II‑induced kidney injury by inactivating nuclear factor‑κB and extracellular signal‑regulated kinase signaling in vivo and in vitro. Int. J. Mol. Med. 48(2), 161 (2021).
  • Zhang, Y., Tan, H., Daniels, J.D., et al. Imidazole ketone erastin induces ferroptosis and slows tumor growth in a mouse lymphoma model. Cell Chem. Biol. 26(5), 623-633 (2019).
  • Zadeh-Ardabili, P.M., Rad, S.K., Rad, S.K., et al. Palm vitamin E reduces locomotor dysfunction and morphological changes induced by spinal cord injury and protects against oxidative damage. Sci. Rep. 7(14365), (2017).
  • Mazereeuw, G., Herrmann, N., Andreazza, A.C., et al. Baseline oxidative stress is associated with memory changes in omega-3 fatty acid treated coronary artery disease patients. Cardiovasc. Psychiatry Neurol. 3674371, (2017). Marottoli, F.M., Trevino, T.N., Geng, X., et al. Autocrine effects of brain endothelial cell-produced human apolipoprotein E on metabolism and inflammation in vitro. Front. Cell Dev. Biol. 9, (2021).
  • Chen, L.-H., Huang, S.-Y., Huang, K.-C., et al. Lactobacillus paracasei PS23 decelerated age-related muscle loss by ensuring mitochondrial function in SAMP8 mice. Aging (Albany NY) 11(2), 756-770 (2019).
  • Wen, J.J., Yin, Y.W., and Garg, N.J. PARP1 depletion improves mitochondrial and heart function in Chagas disease: Effects on POLG dependent mtDNA maintenance. PLoS Pathog. 14(5), e1007065 (2018).
  • Wen, J.J., and Garg, N.J. Manganese superoxide dismutase deficiency exacerbates the mitochondrial ROS production and oxidative damage in Chagas disease. PLoS Negl. Trop. Dis. 12(7), e0006687 (2018).
  • Champion, C.J., and Xu, J. Redox state affects fecundity and insecticide susceptibility in Anopheles gambiae. Scientific Reports 8(13054), (2018).
  • Leonov, A., Feldman, R., Piano, A., et al. Caloric restriction extends yeast chronological lifespan via a mechanism linking cellular aging to cell cycle regulation, maintenance of a quiescent state, entry into a non-quiescent state and survival in the non-quiescent state. Oncotarget 8(41), 69328-69350 (2017).
  • Mandel, E.R., Dunford, E.C., Abdifarkosh, G., et al. The superoxide dismutase mimetic tempol does not alleviate glucocorticoid-mediated rarefaction of rat skeletal muscle capillaries. Physiol. Rep. 5(10), e13243 (2017).
  • Sabow, A.B., Zulkifli, I., Goh, Y.M., et al. Bleedin efficiency, microbiological quality and oxidative stability of meat from goats subjected to slaughter without stunning in comparison with different methods of pre-slaughter electrical stunning. PLoS One 12(5), e0178890 (2017).
  • Adeyemi, K.D., Shittu, R.M., Sabow, A.B., et al. Comparison of myofibrillar protein degradation, antioxidant profile, fatty acids, metmyoglobin reducing activity, physicochemical properties and sensory attributes of gluteus medius and infraspinatus muscles in goats. J. Anim. Sci. Technol. 58(23), (2016).
  • Adeyemi, K.D., Shittu, R.M., Sabow, A.B., et al. Influence of diet and postmortem ageing on oxidative stability of lipids, myoglobin and myofibrillar proteins and quality attributes of Gluteus Medius muscle in goats. PLOS One 11(5), e0154603 (2016).
  • Christensen, L.L., Selman, C., Blount, J.D., et al. Plasma markers of oxidative stress are uncorrelated in a wild mammal. Ecol. Evol. 5(21), 5096-5108 (2015).
  • Adeyemi, K.D., Sabow, A.B., Shittu, R.M., et al. Influence of dietary canola oil and palm oil blend and refrigerated storage on fatty acids, myofibrillar proteins, chemical composition, antioxidant profile and quality attributes of semimembranosus muscle in goats. J. Anim. Sci. Biotechnol. 6(51), (2015).
  • Megson, I.L., Haw, S.J., Newby, D.E., et al. Association between exposure to environmental tobacco smoke and biomarkers of oxidative stress among patients hospitalised with acute myocardial infarction. PLoS One 8(12), e81209 (2013).
  • Nieman, D.C., Gillitt, N.D., Knab, A.M., et al. Influence of a polyphenol-enriched protein powder on exercise-induced inflammation and oxidative stress in athletes: A randomized trial using a metabolomics approah. PLOS one 8(8), e72215 (2013).
  • Xu, Y., Colletier, J.P., Weik, M., et al. Flexibility of aromatic residues in the active-site gorge of acetylcholinesterase: X-ray molecular dynamics. Biophys. J. 95, 2500-2511 (2008).

Furthermore, we also added information about Mycoplasma detection in the method section. We used “Venor GeM Advance Mycoplasma Detection KIT for Conventional PCR” (Cat. No 11-7240; Lott. No. 117S21J2). This kit utilizes the polymerase chain reaction (PCR). For our experiment, we used only Mycoplasma-free cells.

Q3. Please highlight the limitations and implications for researchers carrying out future studies in the same field.

Reply to Q3. As replied to Reviewer 2 the question about long-COVID pathogenesis is completely opened (see reply to Q1 of Reviewer 2). If the persistence of NSPs proteins such as the largest NSP3 is plausible in patients with long-COVID symptoms should be evaluated by proteomic analysis of potential reservoirs in patients. Plasma could be easier to investigate but the analysis of intestinal biopsy in patients that present severe gastrointestinal symptoms associated to long-COVID could confirm our hypothesis and provide a mechanistic explanation of SARS-CoV-2 post-infection sequelae. In this picture, the use of natural molecules (such as HXT) that could controls the expression and the activity of NSP3 may represent a potential preventive approach to reduce the risk of long-COVID development.

Q4. Below, please find other suggestions:

  • KEYWORDS: SARS-CoV-2; C-19???; Hydroxytyrosol; Cytokines; Interferon; Oxidative stress. (The title words should not be repeated in Keywords).
  • INTRODUCTION: Line 69-70: Like As well as for other respiratory viruses, SARS-CoV-2 transmission occurs mainly through the respiratory route. Line 83-84: Lines of evidence suggest that systemic and tissue chronic chronic tissue inflammation, autoimmunity and viral protein/RNA persistence might contribute to the pathogenesis of long-COVID [15-17].
  • DISCUSSION: Line 483-485: On the one hand, Plpro may enzymatically may reduce the ISGylation of MDA5 and IRF3 genes, thus inhibiting the host response to viral infection [30]. Line 511-514: Moreover, several lines of evidence demonstrated that inflammation and oxidative stress are mutually reinforce each other in COVID-19 and presumably also in long-COVID, highlighting the major role of oxidative unbalance as a trigger of both acute and chronic inflammation[27,40]. Line 514-517: Therefore, it is plausible that natural antioxidant molecules may counteract SARS-CoV-2 redox status derangement and consequent inflammation, representing a possible therapy for improve improving signs and symptoms of long-COVID syndrome [26,27,41].
  • CONCLUSIONS: Line 540-542: Many researchers have focused on designing drugs, which can affect replication or protease activity of SARS-CoV-2 to reduce severe/mild COVID-19 disease but, currently, there is an urgent need for requirement of safe and effective treatments to alleviate long-COVID syndrome. Line 548-556: Moreover, we demonstrate that PLpro-dependent adverse effects are reversed by the treatment with HXT, thus indicating the possible effectiveness of this molecule as a longer-term and safe approach to reduce the symptoms of long-COVID in adults, but also in children and adolescents where this condition is a relevant, unrecognized health problem [49].

Reply to Q4.

  • KEYWORDS: We corrected “C-19” with “COVID-19” and SARS-CoV-2 was removed, and Hydroxytyrosol was replaced with Natural antioxidant.
  • INTRODUCTION: We corrected all sentences (Lines 65, 79).
  • DISCUSSION: We revised the discussion and corrected all sentences (Lines 527, 557, 558, 561).
  • CONCLUSION: We corrected all sentences (Lines 590, 599).

Author Response

Reviewer 2

This paper presents the experimental investigation to disclose the mechanism of long-COVID syndrome and the possible therapeutic strategy under the hypothesis that persistent expression of PLpro, one of the NSP of SRAS-CoV-2, might be one of the causes of this syndrome. Most of the performed experiments were well described and done, and the results and the interpretations seem to be consistent and reasonable. This paper seems valuable for the consideration of publication.

Reply to general comment: We thank this Reviewer for appreciating our manuscript and for suggestions that we have follow to improve the manuscript. All changes/addition were highlighted in yellow.

Q1. In the experiment using the forced expression system in vitro, we always have to be cautious for the artificial effects. I would like to know how much expression of PLpro is found in the case of SARS-CoV-2 infection to the culture cells in vitro compared to the forced expression by CMV promotor in this experiment. Also, the authors commented that persistently expressed PLpro might be the background of the long-COVID. Is there any evidence or reference that PLpro is persistently expressed after the infection of SARS-CoV-2 in vitro or in vivo?

Reply to Q1:  We thank the Reviewer 2 for his/her comments. PLpro SARS-CoV2 infected cells was reported in Vero cells [Grenga L, Gallais F, Pible O, Gaillard JC, Gouveia D, Batina H, Bazaline N, Ruat S, Culotta K, Miotello G, Debroas S, Roncato MA, Steinmetz G, Foissard C, Desplan A, Alpha-Bazin B, Almunia C, Gas F, Bellanger L, Armengaud J. Shotgun proteomics analysis of SARS-CoV-2-infected cells and how it can optimize whole viral particle antigen production for vaccines. Emerg Microbes Infect. 2020 Dec;9(1):1712-1721]. The study provide the evidence that the ORF1a papain-like protease (PLpro)/3C-like protease (3CLpro)PLpro is one of the 6 viral proteins found in these cells by proteomic analysis. However, to our knowledge there are no studies reporting the expression of this protein at different MOI. Lacking this reference, we cannot exclude that our forced system is near to a physiological expression of the protein the intro infection models. Regarding mechanisms explaining long-COVID, however, there is no certainty and the hypothesis are different, including persistent viral reservoirs in the body and harmful immune response. Patterson et al. [Patterson BK et al. Persistence of SARS CoV-2 S1 Protein in CD16+ Monocytes in Post-Acute Sequelae of COVID-19 (PASC) up to 15 Months Post-Infection. Front Immunol. 2022 Jan 10;12:746021] showed that viral RNA persistence can occur in monocytes. More importantly, one of the suggested reservoir for SARS-CoV-2 persistence could be de gut. In fact, Zollner et al. [Zollner A, Koch R, Jukic A, Pfister A, Meyer M, Rössler A, Kimpel J, Adolph TE, Tilg H. Postacute COVID-19 is Characterized by Gut Viral Antigen Persistence in Inflammatory Bowel Diseases. Gastroenterology. 2022 May 1:S0016-5085(22)00450-4] found 4 structural viral transcripts in the gut of patients seven months after acute infection. In addition, there are very recent evidence that other body sites could play a role of SARS-Cov-2 reservoir [Chertow, D. et al. Preprint at Research Square https://doi.org/10.21203/rs.3.rs-1139035/v1].

The occurrence of long-term persistence of viral RNA is well-known [Griffin DE. Why does viral RNA sometimes persist after recovery from acute infections? PLoS Biol. 2022 Jun 1;20(6):e3001687], and the hypothesis that not only the viral RNA could persist is known for other RNA viruses. Indeed, non-structural proteins, mainly NSP3, SNP4 and NSP6, may contribute to form double-membrane vesicles where these proteins could be retained and hidden to host cell defence [Santerre M, Arjona SP, Allen CN, Shcherbik N, Sawaya BE. Why do SARS-CoV-2 NSPs rush to the ER? J Neurol. 2021 Jun;268(6):2013-2022]. Therefore, we it is plausible that NSP3 of SARS-CoV-2 could have a similar role in the viral persistency. Further studies are needed to confirm this hypothesis. We added these aspects as a possible limitation of our study in the Discussion section.

See also Reply to Q3 of Reviewer 1.

Q2. The expression of PLpro is reduced by the exposure to 10microM Hydroxytyrosol (HXT). Is the expression of PLpro is also reduced by the exposure to 10microM Hydroxytyrosol when SARS-CoV-2 is infected in the cultured cells? HXT reduces the expression of PLpro and pro-inflammatory cytokines. Is the reduction of pro-inflammatory cytokines directly attributed to the decrease expression of PLpro? Does the antioxidative activity of HXT decrease the expression of pro-inflammatory cytokines? If there are any data available, it is better to present or otherwise at least to comment or argue in the discussion section

Reply to Q2. We thank the Reviewer 2 for his/her comments. Our results demonstrate for the first time that the exogenous expression of the PLpro SARS-CoV-2 NSP3 in Calu-3, Caco-2 and HepG2 cells induces a cascade of inflammatory gene and proteins. Instead, the treatment with 10 uM HXT reduces both the expression of PLpro mRNA and the expression of IFN-related genes and pro-inflammatory cytokine genes/proteins. We hypothesize that the HXT-dependent reduction of inflammation could be mainly due to its capacity in reducing the expression of PLpro. The plausibility of this hypothesis is reinforced by the evidence that HXT per se is not able to affect the cytokine expression as showed in the new Supplementary Figure S4C (see Results and Discussion).

Minor points:

In the abstract, the sentence below is tedious and should be deleted

“Therefore, fully understanding such multifaceted condition, its public health implications and therapeutic management are currently a new priority for the World Health Organization and global scientists”.

As suggested by Reviewer we deleted the sentence.

Round 2

Reviewer 2 Report

In the revised form, the authors responded well and sincerely to my comments.

The revised manuscript seems to be suitable for publication.